# Chemical and Structural Assessment of New Dental Composites with Graphene Exposed to Staining Agents

**DOI:** 10.3390/jfb14030163

**Published:** 2023-03-17

**Authors:** Marioara Moldovan, Diana Dudea, Stanca Cuc, Codruta Sarosi, Doina Prodan, Ioan Petean, Gabriel Furtos, Andrei Ionescu, Nicoleta Ilie

**Affiliations:** 1“Raluca Ripan” Institute of Research in Chemistry, “Babes Bolyai” University, 30 Fantanele Street, 400294 Cluj-Napoca, Romania; 2Department of Prosthetic Dentistry and Dental Materials, Iuliu Hatieganu University of Medicine and Pharmacy, 400006 Cluj-Napoca, Romania; 3Faculty of Chemistry and Chemical Engineering, “Babes-Bolyai” University, 11 Arany Janos Street, 400084 Cluj-Napoca, Romania; 4Oral Microbiology and Biomaterials Laboratory, Department of Biomedical, Surgical and Dental Sciences, Università Degli Studi di Milano, Via Pascal 36, 20133 Milano, Italy; 5Department of Conservative Dentistry and Periodontology, University Hospital, Ludwig-Maximilians-University, Goethestr. 70, 80336 Munich, Germany

**Keywords:** graphene oxide, SiO_2_, ZrO_2_, dental composites, antibacterial test, surface properties

## Abstract

Among the newest trends in dental composites is the use of graphene oxide (GO) nanoparticles to assure better cohesion of the composite and superior properties. Our research used GO to enhance several hydroxyapatite (HA) nanofiller distribution and cohesion in three experimental composites CC, GS, GZ exposed to coffee and red wine staining environments. The presence of silane A-174 on the filler surface was evidenced by FT-IR spectroscopy. Experimental composites were characterized through color stability after 30 days of staining in red wine and coffee, sorption and solubility in distilled water and artificial saliva. Surface properties were measured by optical profilometry and scanning electron microscopy, respectively, and antibacterial properties wer e assessed against *Staphylococcus aureus* and *Escherichia coli*. A colour stability test revealed the best results for GS, followed by GZ, with less stability for CC. Topographical and morphological aspects revealed a synergism between GZ sample nanofiller components that conducted to the lower surface roughness, with less in the GS sample. However, surface roughness variation due to the stain was affected less than colour stability at the macroscopic level. Antibacterial testing revealed good effect against *Staphylococcus aureus* and a moderate effect against *Escherichia coli.*

## 1. Introduction

Restorative materials are being developed continuously by innovative research to improve the predictability of dental treatments. Therefore, more and more complex materials have been developed to facilitate selection of suitable materials and to reduce the number of treatment steps. The development of a palette of universal chromatic composites, is one of these new approaches.

In direct and indirect restorations, determining colour has always been a challenge for dentists. Current technologies for colour determination of restorative composites have been developed to increase the success of colour matching, communication, reproduction and verification in vivo. To increase the efficiency of aesthetic restoration work, it is necessary to objectively quantify the colour of the teeth [1,2], adding high concentrations of micro and nano-powders to provide good strength and wear resistance with unchanged translucency [3,4,5,6], adding radiopacity agents for diagnostics, and adding materials with antibacterial properties [7,8].

Colour stability is a general problem for posterior teeth, but more significant for anterior teeth, which limits the use of composites [9,10]. The evaluation, adoption and implementation of new technologies are important in the identification of composite materials with improved physico-chemical and mechanical characteristics [11,12,13].

In the oral cavity, a discoloration of the surface or substrate of restorations may result due to the degradation, penetration or adsorption of colouring agents at the level of the superficial layer of composite resins. Moreover, surface discoloration can result from surface integrity, roughness and the polishing technique [14].

Most studies regarding the colour stability of dental restorative materials, glass ionomer cements, compomers, and indirect composites, have investigated the effects of tea and coffee. The results of these studies have shown that the biggest colour and shade changes occur using tea [15], but others have indicated that coffee has the biggest effect in changing colour [16].

Micro-structured elements within coffee can be adsorbed onto the enamel surface and generate dirt clusters [17,18]. Data in the literature shows that red wine also has great staining potential due to chromophore particle adhesion onto solid surfaces [19,20]. Deposits might affect surface nanostructure of enamel and dental materials, generating local deposit crusts [21,22].

Filler materials play an important role in composite resistance to various stains such as coffee and red wine. Hydroxyapatite nanoparticles are similar to the basic structures in the enamel and have poor anti-staining properties. However, addition of mineral enhancers could improve the anti-staining effect. Thus, silicon dioxide SiO_2_ is used as an additive to improve hydroxyapatite behaviour [23,24], as is zirconium dioxide [25]. Graphene oxide is another enhancer in dental composites is [26,27,28] and moderates mineral filler nanoparticles distribution in the matrix, and also acts as bioactive compound in dental cements [29].

As a result of the development of nanotechnology, a wide range of materials have emerged with excellent durability and handling possibilities, improved bonding ability, lower shrinkage, improved finishing, and aesthetically improved optical effects. The most important factor in achieving the objectives of restorations is the clinicians’ knowledge of the nuances of the material and the techniques they work with.

The aim of present research was to investigate a new dental composite based on graphene oxide in the presence of different hydroxyapatite: pure HA, HA-SiO_2_ and HA-ZrO_2_. The null hypothesis is that the graphene oxide addition would have no effect on composite surface staining by coffee or red wine.

The novelty of our study relates to the fillers used in the composition of dental composites to confer colour stability and antibacterial properties.

## 2. Materials and Methods

### 2.1. Preparation of GO/SiO_2_ and GO/ZrO_2_ Materials

SiO_2_ powder (Aerosil 200; Degussa, Germany), ZrO_2_ (Merck Schuchardt OHG, Hohenbrunn, Germany) and graphene oxide (GO) were used as starting materials. Graphene oxide (GO) was prepared from natural graphite using a modified Hummers’ method [30]. GO/SiO_2_ were prepared by mixing GO and SiO_2_, followed by sonication for 45 min and drying at 50 °C in an oven, as described in previous research [31].

GO/ZrO_2_ was synthetized by mixing two suspensions: GO (7.5 mg) dispersed in double distilled water (10 mL) and ZrO_2_ (100 mg) dispersed in NaOH (15 mL, pH 9.5). Powders were separately dispersed in an aqueous system for 15 min by ultrasound waves. The GO/ZrO_2_ suspensions were mixing using ultrasound waves for another 45 min. Then, the resultant mixtures were dried at 50 °C in an oven. The obtained powder was ground in a mill to produce uniform granulation.

### 2.2. Composite Formulation

The composites hereinafter referred to as GS and GZ were prepared as monopastes by dispersing GO/ZrO_2_ and GO/SiO_2_ powders in a matrix comprising four distinct components. The complex mixtures were composed of a light-curing resin (forming a hard material as a result of cross-linking when exposed to light), inorganic filler (improving properties of the polymers, such as tensile and compressive strength, abrasion resistance, and thermal stability), a coupling agent (enhancing filler/resin interactions) and an initiator/accelerator system (activating the photo-polymerization reaction). The procedures were conducted according to information in the literature [32,33].

The organic matrix (polymerizable resin) consisted of an aromatic base monomer, bisphenol-A glycidyl dimethacrylate (Bis-GMA), combined with a low-molecular weight diluent monomer, triethylenglycol-dimethacrylate (TEGDMA), that enhanced molecular mobility in the polymerization process. The composites contained fillers such as quartz (Uricani, Romania), nanoparticles as a reinforcing constituent, and bioglass with strontium/zirconium and barium that increased radiopacity [30]. Composite compositions are presented in Table 1.

Bis-GMA, HA-SiO_2_, HA-ZrO_2_ nanoparticles, as well as the glass-ceramics SiO_2_-SrO-ZrO_2_-Al_2_O_3_-B_2_O_3_-NaF-CaF (35-20-10-10-13-6-6 wt%) were synthetized in our laboratory (UBB-ICCRR, Babes-Bolyai University, Raluca Ripan Institute of Research in Chemistry). TEGDMA was purchased from Sigma Aldrich, Darmstadt, Germany. The filler surface particles were modified by using γ-methacriloyloxypropyl-trymethoxysilane (A-174) (Sigma Aldrich, Germany) to provide an improved interaction between the filler and polymer matrix.

The light-curing process of composites was performed using a Woodpecker^®^ Dental Curing Light 1400 mW/cm^2^ LED.B lamp for 80 s, and an activating photo-initiator/accelerator system, i.e., camphorquinone, CQ (Sigma Aldrich, Germany)/dimethyl (aminoethyl) methacrylate, DMAEM (Sigma Aldrich, Germany). A sample denoted CC was considered the control specimen.

### 2.3. Investigation Methodology

#### 2.3.1. FT-IR Spectroscopy

The synthesized glasses were characterized by an FTIR spectrophotometer JASCO—610 (JASCO International Co., Ltd., Tokyo, Japan) in the range of 400–4000 cm^−1^ using a pelletizing technique in KBr. For silanization, we used hydrolyzed silane A-174 in an acidified alcoholic solution at pH 3.5–4. The silanized powder was initially washed with acetone to remove free adsorbed silane, and that remaining after washing from the deposited silane was chemisorbed.

#### 2.3.2. Evaluation of Composite Colour Stability

Fifteen disc-shaped 1mm × 15 mm specimens of each composite were prepared using Teflon moulds and irradiated with a visible light-curing unit (Wodpecker LED lamp) for 20 s/point at nine different points on both sides. Samples were divided into three groups of five and were immersed in two different staining solutions (coffee or red wine) for 1 h/day for 30 days. The coffee solution (pH = 5) was prepared using 5 g coffee (Jacobs Krönung, Bremen, Germany) in 200 mL of boiled distilled water. Red wine from our region, Issa Pinot Noir (La Salina, Turda, Romania), 200 mL, 13.5% alcohol, pH between 2.9 and 3.9, was used without dilution. Both colour solutions were changed every day. A colorimetric evaluation was performed on a white background using a VİTA Easyshade Compact (VİTA Zahnfabrik, Germany) dental spectrophotometer. Samples were cleaned with deionized water and air-dried before colour measurement. The colour change was identified using the CIE *L*, a*, b** system with three-dimensional representation of *L*, a*, b** chromaticity space. *L** represents the white/black coordinate, *a** indicates the red/green coordinate and *b** the yellow/blue coordinate. The total color difference between all three coordinates was calculated using the equation:Δ*E* = [(Δ*L**)^2^ + (Δ*a**)^2^ + (Δ*b**)^2^]^½^,(1)
where Δ*L**, Δ*a**, and Δ*b**are the differences in the *L*, a** and *b** colour parameters between the two colours [34,35].

Colour changes (*ΔE*) were interpreted in NBS (National Bureau of Standards) units, with the formula NBS units = *∆E** × 0.92. The clinical relevance of the data was interpreted according to the NBS criteria, namely: trace—very slight color change (0–0.5); light—slight color shift (0.5–1.5); visible—perceptible change (1.5–3.0); appreciable—large color change (3.0–6.0); very—greater color change (6.0–12.0) and excessive—change to another color (12.0+).

#### 2.3.3. Sorption and Solubility Test

Ten specimens were prepared, similar to those used for colour stability evaluation, and divided into two groups (five for each material). Each group was immersed for 28 days at 37 °C in distilled water and artificial saliva. Sorption and solubility measurements were performed according to ADA Specification No. 27-1993/ISO 4049/2000 regarding Polymer-based filling, restorative and luting materials [36,37].

The specimens were weighted at 7, 14, 21 and 28 days after removing from the medium and placed in a desiccators containing dried silica gel until a constant mass was measured.

Sorption (W_SP_) and solubility (W_SL_) values (in μg/mm^3^) were calculated using Formulas (2) and (3):W_SP_ = (m_1_ − m_2_)/V(2)
W_SL_= (m_0_ − m_2_)/V(3)
where m_0_—samples weight before immersion (μg), m_1_—samples weight after immersion (μg), m_2_—samples weight after desiccation (μg), and V—sample volume (mm^3^).

#### 2.3.4. Surface Optical Profilometry

Sample topography and surface morphology were investigated with a Ze Gage Zygo Ametek laser optical profiler system (Middlefield, CT, USA). The surfaces were scanned in non-contact mode with a scanned area of 400 µm × 400 µm. Each recorded profile was investigated using Zygo analysis software, and surface roughness was measured using Ra and Rq parameters [15]. The maximum surface height was also recorded.

The resulting profiles were further processed using Image J 1.53k microscopic specialized software using a monochrome cyan outfit for better expression of the topographical images. The darkest spot represented the lowest point of the surface and the brightest pixel represented the highest point of the surface. A three-dimensional profile was presented below each topographic image featuring the absolute value of the surface height on the *Z*-axis.

#### 2.3.5. Scanning Electron Microscopy (SEM)

Composite samples were investigated with a scanning electron microscope (SEM; Inspect™ S produced by FEI Company, Hillsboro, OR, USA). The samples were investigated in low vacuum mode to assure an optimal view of the surface details at an acceleration voltage of 25.00 kV. The samples were not gold coated to assure better individualization of the filler particles in connection with the polymer matrix. Each sample was inspected in three different macroscopic areas.

#### 2.3.6. Antibacterial Activity

In vitro antibacterial activities of composites were determined by agar paper disc diffusion and well diffusion methods. Two pathogenic strains including *E. coli* ATCC 25922 and *S. aureus* ATCC 25923 as Gram-negative and Gram-positive bacteria (Microbiology laboratory of Bio-Labs, Frankfurt, Germany) were used.

The filter paper disks (5 mm in diameter) were separately impregnated with CC, GS and GZ composites and placed on the surface of an agar plate previously inoculated with bacterial strains. Plates were incubated at 37 °C for 48 h. Then, the diameter of the clear zones around each disc (inhibition zone) was measured in millimetres.

A similar procedure was used in the agar well diffusion assay. The agar plate surface was inoculated with selected bacterial strains. Wells 8 mm in diameter were made, and 20 μL of each composite was dispensed into the wells. Then, the agar plates were incubated under the same conditions described above. Inhibition of the bacterial growth was measured using a scale in millimetres. The interpretation of antibacterial activity of GS and GZ composites was evaluated by comparing the results to a standard zone size. Tests were performed in triplicate to minimize errors.

#### 2.3.7. Statistical Analysis

Roughness parameter data were analysed statistically using repeated measurements analysis of variance (RM-ANOVA). Student’s *t*-test was used to analyse roughness, colour differences (ΔE) and ΔL*, Δa*, and Δb* between groups. Statistical significance was established for all tests at the probability value of *p* < 0.05.

Measurements were repeated at least three times, and data represented as mean and standard deviation (SD).

For all statistical tests, a result was considered statistically significant at p = 0.05. Statistical calculations were performed using Microsoft Excel (version 7).

## 3. Results

### 3.1. FT-IR Spectroscopy Analysis

From the FT-IR spectra, information was obtained about the structure of the inorganic filler used in the formulation of dental composites, as well as the interfacial connection between silane A-174 and the surface of the synthesized inorganic fillers.

Figure 1 shows the spectra of the S1, S2, non-silanized powders compared to the silanized powders, as well as the A-174 silane spectra. The presence of silane on the glass surfaces and peaks specific to silane could be observed in the spectra.

### 3.2. Colour Stability of Composites

The differences between the specimens for different time of immersion (7 and 30 days) with respect to the mean Δ*E* values and NBS units with corresponding standard deviation (SD) are presented in Figure 2. Larger colour changes occurred in control specimen CC both in coffee and wine after 30 days (NBS > 4). No noteworthy colour changes were observed for the composite containing GO/SiO_2_ after immersion in either staining solution. However, the GZ specimen showed higher ΔE values than those of GS, as well as a larger variation of colour alteration values. According to the NBS system, “slight colour change” in the GS specimen was observed after 30 days of immersion in the coffee solution. The trend was similar for the GS sample using red wine as staining solution after 7 days, with a slight shift towards a “perceivable colour change” at the end of 30 days. The GZ specimen showed NBS units that were almost two times higher than those of the GS specimen in the same colour media, except for the value after 30 days of immersion in red wine. The experimental results indicate a noticeable colour change of this composite in both staining solutions for the entire immersion period in the clinical environment, there were acceptable colour changes of the composite containing GO/SiO_2_, while the composite with GO/ZrO_2_ was classified as showing “noticeable/appreciable colour change”.

Statistically significant differences were observed between all the composites tested for all the monitored parameters (*p* < 0.001). However, following the analysis of multiple comparisons, it was found that only CC differed significantly from the other tested substances (*p* < 0.001).

### 3.3. Sorption and Solubility Test

The solubility test was conducted according to ISO 4049 [36] requirements. The maximum values of water sorption (W_SP_) were below 40 μg/mm^3^ and water solubility (WSL) did not exceed 7.5 μg/mm^3^ for the specified materials. The result of sorption and solubility in distilled water and artificial saliva are presented in Figure 3.

The variations in Figure 3a shows that the control sample had progressively increasing water sorption with exposure time, proving that simple hydroxyapatite is relatively permeable to water and allows relatively high infiltration. The composite mixture containing silicon dioxide and graphene proved to be more resistant to water sorption up to 20 days of exposure, after which solubility increased. This is probably about of the water infiltration beside the shield formed by the SiO_2_ nanoparticles mixed up with graphene nano foils.

Water exposure of the GZ sample caused a significant increasing of the sorption up to 5 days, after which it significantly decreased. The absorbed water might influence upper layer interaction with ZrO_2_ activators of hydroxyapatite to act synergistically with the graphene nano-foil to prevent further liquid infiltration. All composite samples had some solubility on contact with water but that decreased significantly with exposure time (Figure 3b). This is probably due to the optimal cohesion between filler particles within the polymer rather than the particular enhancement of hydroxyapatite.

Exposure to artificial saliva revealed relative water absorption for GS and GZ samples, and stationary behaviour for the control sample up to 5 days (Figure 3c). Solubility in artificial saliva (Figure 3d) was similar to that observed for water exposure. The solubility of materials was progressive, with negative values explained by the removal of residual monomer from the surface of the material and the reduced weight of the specimens. Experimental composites with higher percentages of graphene had significantly higher solubility and water sorption values after water immersion (*p* < 0.001).

### 3.4. Optical Profilometry

The topography of the CC surface sample is shown in optical profile in Figure 4a. The high filler amount (i.e., 80%) within the Bis-GMA/TEGDMA matrix generated a dense composite, which influenced surface morphology. Local heights of about 8.86 μm occurred due to the strong compaction between the polymer mixture and filler particles in clusters with 30–50 μm diameters. The less compact areas generated several topographic depressions with rounded aspects and dendritic borders, having diameters in the range of 30–80 μm. The resulting mean roughness values (Ra = 93.36 nm; Rq = 152.84 nm) were considered the standard reference for this composition.

The optical profile in Figure 4b shows the CC sample surface topography completely filled by coffee deposits. This caused a significant decrease in roughness but significant dirt clusters occurred. The profile has dendritic shape with a rounded core of about 30–50 μm and dendritic whiskers of about 100 μm length and less than 10 μm in diameter. Therefore, the mean roughness values significantly increased, with Ra = 224.30 nm and Rq = 402.59 nm. The surface height increased to 11.17 μm.

CC sample exposed to red wine showed altered topography (Figure 4c). The topographic features eroded by the acid component of the wine caused local delamination of filler particles within the topographic depressions. This caused mineral loss and large increasing in mean roughness means of Ra = 1244.40 nm and Rq = 1893.57 nm. The variation in Figure 5 shows that red wine had the most erosive effect on CC sample topography.

The topography of untreated the GS sample (Figure 4d) was very compact and uniform. The mineral filler to Bis-GMA/TEGDMA polymer matrix ration was the same as in the CC sample (80/20). Thus, the difference was caused by the addition of GO/SiO_2_, which facilitated better mixing of the mineral particles with the polymer. Several small spots observed on the surface related to the Sr-Zr glass and quartz particles. The mean roughness values were the lowest among the samples investigated in the current research (Ra = 53.1 nm and Rq = 124.58 nm). There were no open pores filled by microscopic coffee particles facilitating dirt cluster formation directly on the filler particle spots, which act as coalescence centres (Figure 4e). The local height increased greatly to 15.81 μm from only 1.28 μm observed for the initial sample. The mean roughness values consequently increased, as shown in Figure 4 (Ra = 282.62 nm and Rq = 680.38 nm).

GS sample exposure to red wine led to a significant topographical change (Figure 4f). A lack of cohesion between filler particles and the polymer matrix occurred with significant mineral loss. Therefore, the local height increased to 20.31 μm and the roughness values were highest among all samples investigated (Ra = 1619.68 nm and Rq = 2057.28 nm). Physicochemical weakness between the filler and matrix caused infiltration of the red wine.

The topography of the initial GZ sample (Figure 4g) was quite compact without open pores and with good cohesion between the mineral filler and polymer matrix. This proves that addition of GO/ZrO_2_ to the initial composition facilitates compactness of the microstructure. However, some lower plate areas with a dendritic shape were observed with local heights of about 3.81 μm. Therefore, the average roughness levels (Ra = 63.82 nm and Rq = 253.65 nm) were lower than those of the CC initial sample but higher than those of the GS sample.

The topography of the GZ sample exposed to coffee (Figure 4h) was very similar to that of the unstained sample. Only a few small dirt deposits were observed on top, causing local heights to increase slightly to 5.45 μm. The mean values of roughness were Ra = 118.46 and Rq = 309.35 nm, which were the lowest values among the coffee-stained samples (Figure 5). This shows that GO/ZrO_2_ additions made the composition resistant to coffee micro-particles deposition.

GZ sample exposure to red wine resulted in well preserved microstructure, but several small, rounded pores occurred due to acid interaction with exposed filler particles within the outer most layers (Figure 4i). Pore diameters ranged between 5–25 μm. Mean roughness was influenced as a consequence, with Ra = 910.27 and Rq = 1300.99. These values were the highest among the GZ samples, and the lowest among the samples exposed to red wine, showing that GO/ZrO_2_ addition improved composite resistance against the erosive effect of red wine.

### 3.5. SEM Investigation of Samples

The optical profiles in Figure 4 reveal with high accuracy the topography of the investigated samples but do not allow detailed observation of microstructure evolution. SEM microscopy was required for this, and the obtained SEM images are presented in Figure 6.

Figure 6a show the top of a CC initial sample topographic cluster revealing its microstructure. There appear to be Sr-Zr particles within boulder shape with diameters of about 5 μm, and fine quartz particles of about 1–3 μm in diameter that are embedded into the polymer matrix. The HA-Ag particles have submicron dimensions and well dispersed among larger filler particles, assuring optimal compaction of the composite. Coffee exposure affected the CC microstructure, with several material deposits placed mainly on the bigger filler particles but also covering significant areas of the submicron particles (Figure 6b). These morphological alterations influenced the aesthetical aspects of the CC samples but did not affect composite cohesion. Red wine action proved to be more aggressive than coffee (Figure 6c), and surface mineral loss occurred, as shown by several depressions with boulder-like margins, and sizes in the range of 3–5 μm. These morphological changes suggest that wine infiltration between Sr-Zr glass particles and polymer caused local delamination, which facilitated their loss.

The morphology of the GS sample was very compact (Figure 6d) and the addition of GO/SiO_2_ enhanced filler particle embedding in the polymer matrix, in good agreement with optical profilometry observation. Submicron HA-Ag had greater interaction with the Sr-Zr and quartz filler particles due to the graphene compound interlocking with the microstructure. Coffee exposure did not affect GS sample microstructure cohesion but generated a significant, consistent dirt film deposit (Figure 6e)The situation was significantly changed after the red wine exposure (Figure 6f). The observed large eroded areas imply significant mineral filler loss.

The addition of GO/ZrO_2_ to the initial composition improved GZ sample microstructure cohesion, as seen in Figure 6g, which shows coffee exposure resulted in microstructural dirt deposits in certain areas such as the centre and the right side of the image (Figure 6h). Figure 6i shows good resistance of the GZ composite against acid erosion induced by the red wine. Only a few fine particles from the surface were etched by wine exposure, bigger particles being more evident, but the cohesion within the microstructural components and polymer matrix was not affected.

### 3.6. Antibacterial Activity

The results of monitoring antibacterial activity of the investigated composites are presented in Figure 7, and the inhibition zone measurements are represented in Figure 8.

After incubation at 37 °C for 48 h, the zone of inhibition around the powders was measured. S inhibition was observed of the Gram-positive bacterium *S. aureus*.

Figure 7 shows that the composite sample GS, with GO/SiO_2_ composition, produced the largest zone of inhibition against both *S. aureus* and *E.coli*. However, Gram-negative bacteria (*E. coli*) were less sensitive than Gram-positive (*S. aureus*) to all samples containing GO/SiO_2_. In the case of samples GZ with GO/ZrO_2_, a smaller inhibition zone was observed.

The observed antibacterial inhibition supports the idea that composites based on graphene oxide with various oxides may have intense antibacterial effects against the tested microorganisms.

## 4. Discussion

The experimental materials analysed in this study extend modern research trends in the development of dental materials.

In silicate glasses, the silicon is usually tetracoordinated by oxygen. The structure of silicate glasses can be described as a mixture of units or structural entities of interconnected SiO_4_ and AlO_4_ tetrahedra. In the glass synthesis, we started from SiO_2_ whose IR spectrum is dominated by the 1106 cm^−1^ band (a shoulder at 1168 cm^−1^) which can be attributed to stretching vibrations υ(Si–O–Si). The bands at ~800 cm^−1^ are attributed to the deformation vibrations δ(O –Si–O), and the bands at ~475 cm^−1^ are attributed to the deformation vibration δ(Si-O-Si). From the spectra, bands were observed that suggest the presence of an organic residue of the methacrylic group, given the presence of a band due to in-plane deformation vibration at 1453 cm^−1^ and a band at 1638 cm^−1^ corresponding to the elongation of the υ(C=C) bond from the methacrylic group, as well as an intense absorption band at 1720 cm^−1^ attributed to the bond υ(C=O) in the methacrylic group in the silane structure. Bands in the range 2800–3000 cm^−1^ are due to asymmetric and symmetric vibrations of the CH_3_ and CH_2_ groups.

In the range 1000–1200 cm^−1^, the formation of new types of Si-O bonds was indicated following silanization of the filler. The increase in intensity of the band due to the deformation vibration of the Si-O-Si bond, δ(Si–O–Si), at 470 cm^−1^, from the non-silanized powder, indicates an increase in the number of Si_2_O_5_ units and SiO_3_ chains with oxygen, which were not in the deck. The presence of absorption bands characteristic of the silane structure in the IR spectrum confirms that the silane is deposited on the powder and is chemically bound.

The colour stability of composites can be affected by various factors such as the resin matrix, filler particle size, photo-polymerization conditions, staining agents, and staining solution pH values, among others [38,39]. The resin composite could be degraded by water sorption as a result of exposure to saliva media, thus weakening resin-filler interface bonding and promoting micro cracks, thereby affecting aesthetics by stain penetration and discoloration [40].

Large filler particles cause rougher surfaces than smaller ones, leading to higher staining susceptibility and more colour permeability [41]. Photo-polymerization is a complex process influenced by many factors including resin composite shade, resin composite increment thickness, light curing duration, the light curing system, light curing tip distance from resin composite surface, light curing duration, and filler type [42].

The colour stability of resin composite has been evaluated after exposure to a variety of staining media, especially coffee, red and white wine, cola, juices, tea and whisky, as the main staining agents that could change the colour of dental composites and teeth [39]. Wine affects the structure of resin composites and teeth due to its alcohol content, which causes increased surface roughness and erosion in composites [19,20]. Lower pH values of coloured media has been reported to increase staining, possible by favouring degradation of the polymer chain [43,44]. Some studies have reported that colour changes are influenced by increased surface roughness allowing stain penetration [39]. By contrast, other studies have shown no correlation between surface irregularities and colour changes in restorative materials [45]. In accordance with with these findings, it was concluded that our results are in accordance with previous studies, and the composites had similar colour parameters [46,47].

A general tendency of solubility decreasing was observed from 5 to 20 days of exposure, which might be explained by relative crystallization of saliva minerals at the contact with filler nanoparticles.

Coffee is a very popular drink that contains microscopic, ground coffee bean particles. These fine particles move into the liquid under Brownian motion, and tend to adsorb onto material surfaces wetted by coffee flow [15,17]. Data in the literature show that microscopic coffee particles fill cavities in rough surfaces with open pores (e.g., zirconia ceramics [17,48] and teeth enamel surfaces [14,15] with a tendency to form dirt clusters.

The observations made on our samples confirmed previously published data, in that Suriyasangpetch et al. [49] showed that surface roughness was related to red wine staining, and Lee et al. [22] related red wine stain persistence to the roughness of aesthetic CAD-CAM restorations. SEM microscopic investigation was required to investigate the dissolution mechanism. It is possible that wine infiltration is caused by the silicate content within the GO/SiO_2_ complex. Particles from the GS composite were rounded to dendritic, and sizes ranged from about 5 μm up to 30 μm, in good agreement with optical profilometry observations. Coffee exposure did not affect GS sample microstructure cohesion but generate a significant and consistent dirt film deposit, while red wine facilitated delamination of bigger filler particles in the presence of GO/SiO_2_.

The synergy between graphene and zirconia-based composites ensures optimal compaction of the composite. The presence of GO/ZrO_2_ nanoparticles in the GZ composite results in greater stability of the material against acid erosion induced by red wine. Exposure to coffee covered certain areas with microstructural dirt deposits.

## 5. Conclusions

The colour stability of resin composites evaluated after exposure to coffee and red wine revealed less colour changes of the composite containing GO/SiO_2_ compared with the composite with GO/ZrO_2_, which showed appreciable colour change.

A general tendency of decreased solubility was observed from 5 to 20 days of exposure, which might be explained by relative crystallization of saliva minerals at the contact with filler nanoparticles. Solubility in artificial saliva was similar with that observed for the water exposure.

Optical laser profilometry of the samples showed that the addition of GO/ZrO_2_ improved composite resistance to discoloration by coffee and the erosive effect of red wine, whereas the addition of GO/SiO_2_ weakened composite resistance against the effects of coffee and red wine.

Antibacterial testing revealed a good effect against *Staphylococcus aureus* and a moderate effect against *Escherichia coli*.

## Figures and Tables

**Figure 1 jfb-14-00163-f001:**
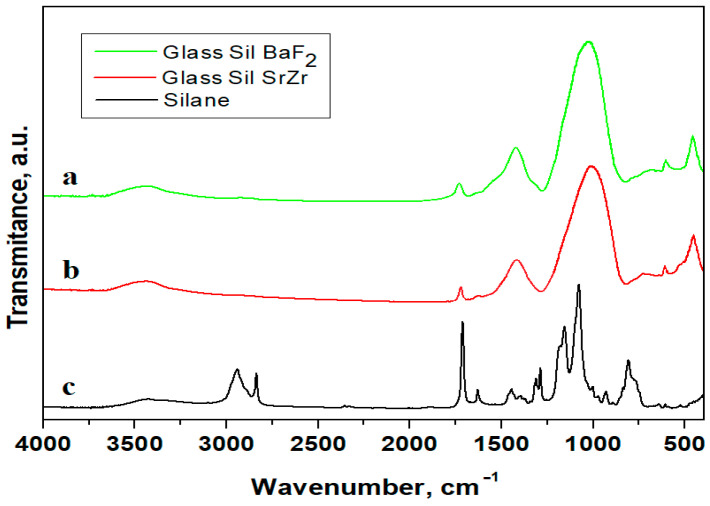
FTIR spectra for (**a**) silanized glass with BaF_2_, (**b**) silanized glass with Sr and Zr and (**c**) A-174 Silane.

**Figure 2 jfb-14-00163-f002:**
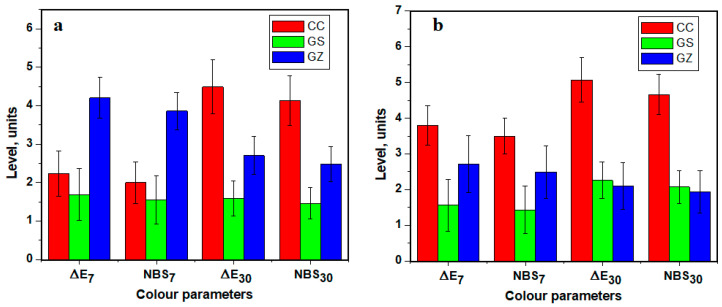
ΔE and NBS (mean ± SD) values of all specimens at 7 days and 30 days of immersion: (**a**) coffee, and (**b**) red wine.

**Figure 3 jfb-14-00163-f003:**
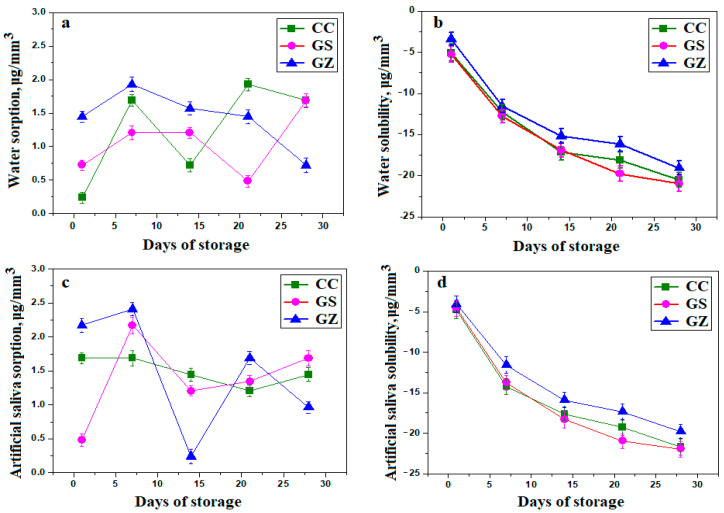
Mean values and standard deviation for (**a**) water sorption (μg/mm^3^), (**b**) water solubility (μg/mm^3^), (**c**) artificial saliva sorption (μg/mm^3^) and (**d**) water solubility (μg/mm^3^) of the composite materials tested after 1, 7, 14, 21 and 28 days of artificial saliva storage.

**Figure 4 jfb-14-00163-f004:**
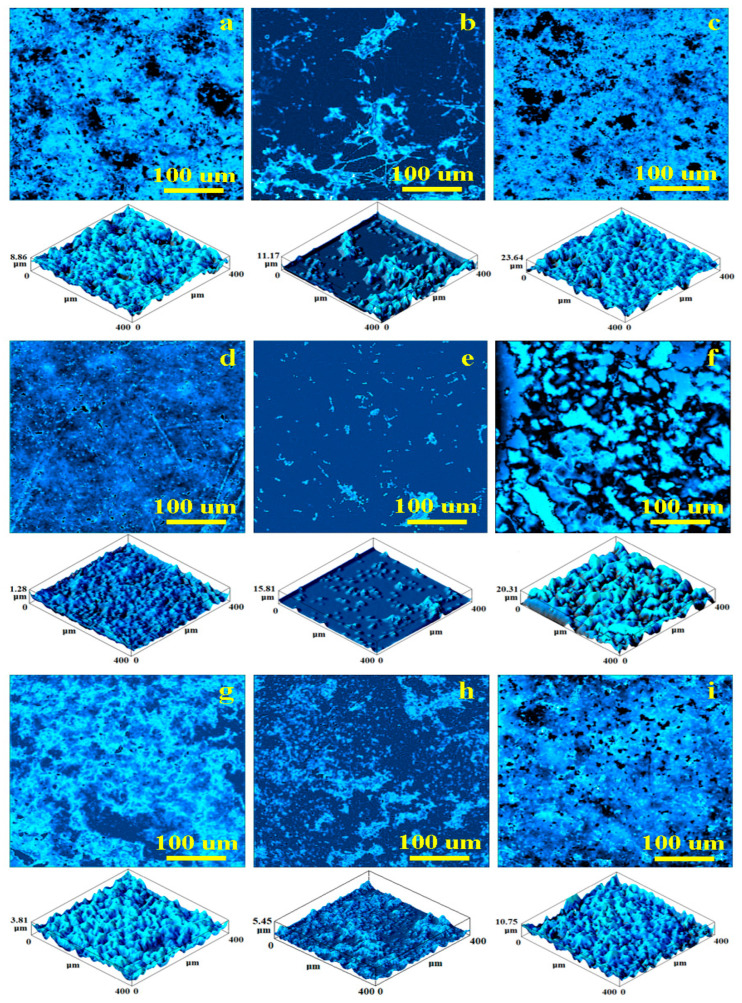
Optical profiles for the investigated samples: (**a**) CC initial, (**b**) CC coffee stained, (**c**) CC red wine stained, (**d**) GS initial, (**e**) GS coffee stained, (**f**) GS red wine stained, (**g**) GZ initial, (**h**) GZ coffee stained, and (**i**) GZ red wine stained.

**Figure 5 jfb-14-00163-f005:**
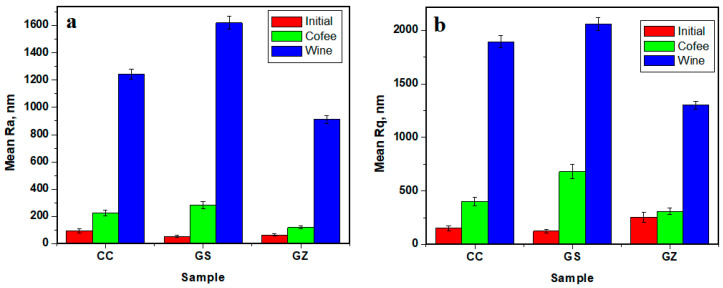
Roughness variation as a function of the graphene content and exposure environment: (**a**) Ra, and (**b**) Rq.

**Figure 6 jfb-14-00163-f006:**
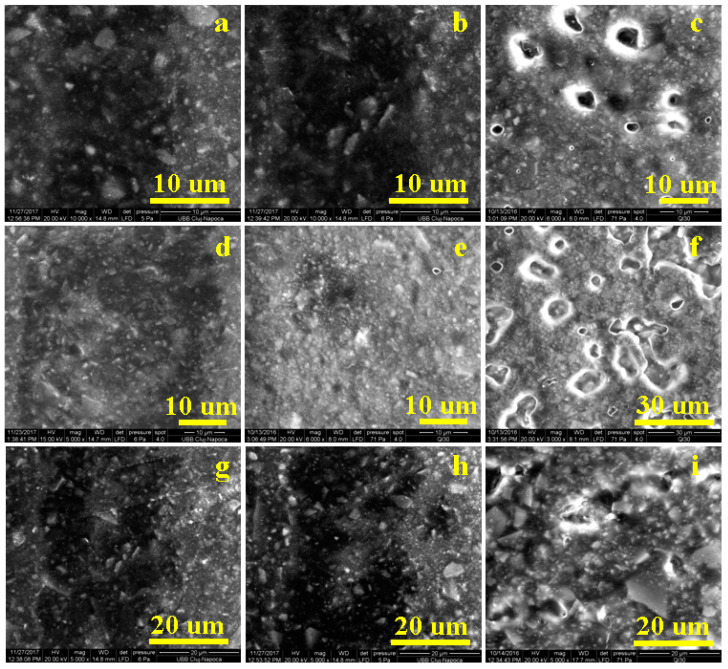
SEM images of the investigated samples: (**a**) CC initial, (**b**) CC coffee stained, (**c**) CC red wine stained, (**d**) GS initial, (**e**) GS coffee stained, (**f**) GS red wine stained, (**g**) GZ initial, (**h**) GZ coffee stained, and (**i**) GZ red wine stained.

**Figure 7 jfb-14-00163-f007:**
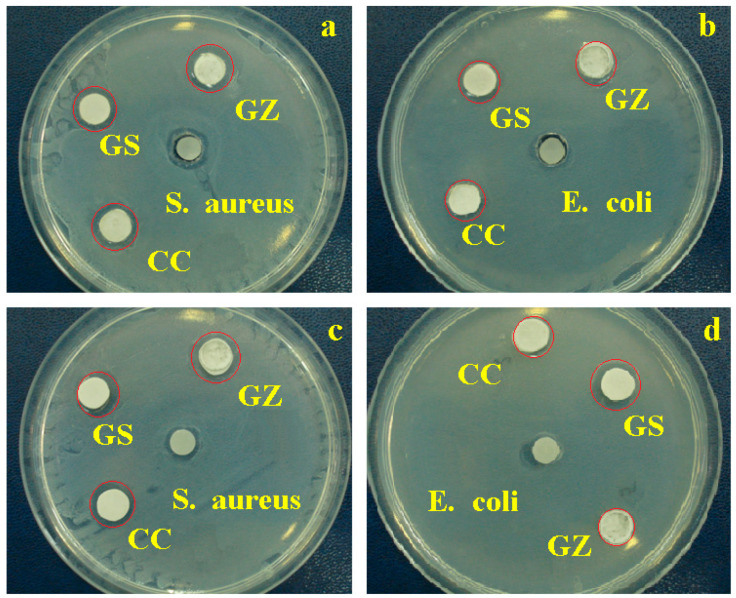
Antibacterial activity tested by agar paper disk diffusion: (**a**) *Staphylococcus aureus*, (**b**) *Escherichia coli*; and agar well diffusion (**c**) *Staphylococcus aureus*, (**d**) *Escherichia coli*.

**Figure 8 jfb-14-00163-f008:**
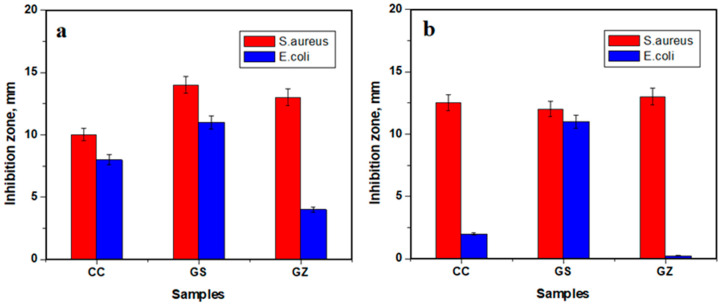
Inhibition zones resulting from antibacterial activity tested using (**a**) agar paper disk diffusion, and (**b**) agar well diffusion.

**Table 1 jfb-14-00163-t001:** Composite compositions.

Type	Materials	Manufacturer	Monomers Matrix 20 wt%	Total Fillercontent 80 wt%
light-curing composite	CC	UBB-ICCRR, Cluj-Napoca Romania	Bis-GMA; TEGDMA	HA-Ag(particle size 0.01–60 μm and 5–8 nm); Sr-Zr glass (particle size 0.01–0.035 μm and 2–6 nm); Quartz
light-curing composite	GS	UBB-ICCRR, Cluj-Napoca Romania	Bis-GMA; TEGDMA	0.3 GO/SiO_2_ HA-SiO_2_(particle size 0.01–60 μm and 5–8 nm); Sr-Zr glass (particle size 0.01–0.035 μm and 2–6 nm); Quartz
light-curing composite	GZ	UBB-ICCRR, Cluj-Napoca Romania	Bis-GMA; TEGDMA	0.3 GO/ZrO_2_ HA-ZrO_2_ (particle size 0.01–60 μm and 5–8 nm); silica; barium glass (BaO) (particle size 0.01-0.035 μm and 2–6 nm); quartz;;silica; glass filler (with BaF_2_) (size 2–6 nm)

## Data Availability

Not applicable.

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
