# Peer review of "Chemical and Structural Assessment of New Dental Composites with Graphene Exposed to Staining Agents"

_jfb, 2023, doi:10.3390/jfb14030163_

Round 1

Reviewer 1 Report

Authors should clearly state the added value of the work, compared with what is already puvlished in the literature, at the end of the "Introdution" section.

I could not follow the CC samples. They are pure HA or HA-Ag? The two terms appear in the manuscript, and there is no explanation for this.

The GO composites contain pure HA or HA-Ag? Please, clarify this.

Author Response

Dear reviewer, 

The authors consider the reviewer’s comments and suggestions of highly scientific importance. Your ideas really helped us to improve the quality of our manuscript. We have corrected the errors that appeared in the manuscript and incorporated the changes in the revised manuscript using track changes.

Authors should clearly state the added value of the work, compared with what is already published in the literature, at the end of the "Introdution" section.

Response:  We added in the introduction the novelty of the study.

I could not follow the CC samples. They are pure HA or HA-Ag? The two terms appear in the manuscript, and there is no explanation for this.

Response: The GO composites contain pure HA. HA-Ag are presented only in CC composite.

Reviewer 2 Report

This is a sound paper that reports the effect of adding graphene oxide in various forms to experimental dental composite resins.  Colour stability against coffee and red wine, and antimicrobial effects were studied.  The resulting paper is worthwhile and shows that the best resistance to colour change was provided by the graphene oxide/zirconia additive.

A few changes are needed.

Line 45: Colour stability in posterior teeth is not a problem because the teeth can hardly be seen.  This section should be rewritten to state that colour stability of composites is a general problem, and more significant in the anterior teeth.

Line 34 (and elsewhere): Names of bacteria should have a lower case letter for the second part of the same, i.e. Staphylococcus aureus.  This also needs changing on Figure 5, among other places.

Line 451: The word "who" should be "which".

Author Response

Dear reviewer,

The authors consider the reviewer’s comments and suggestions of highly scientific importance. Your ideas really helped us to improve the quality of our manuscript. We have corrected the errors that appeared in the manuscript and incorporated the changes in the revised manuscript using track changes.

This is a sound paper that reports the effect of adding graphene oxide in various forms to experimental dental composite resins.  Colour stability against coffee and red wine, and antimicrobial effects were studied.  The resulting paper is worthwhile and shows that the best resistance to colour change was provided by the graphene oxide/zirconia additive.

A few changes are needed.

Line 45: Colour stability in posterior teeth is not a problem because the teeth can hardly be seen.  This section should be rewritten to state that colour stability of composites is a general problem, and more significant in the anterior teeth.

Response: Paragraph Line 45 was improved according to the recommendations: ,, However, there are problems which limit the use of composites, especially in posterior teeth but the composite colour stability is a general problem that is more significant in the anterior teeth. [9, 10].”

Line 34 (and elsewhere): Names of bacteria should have a lower case letter for the second part of the same, i.e. Staphylococcus aureus.  This also needs changing on Figure 5, among other places.

Response: The manuscript was revised according to the recommendation and ,,Figure 5.Antibacterial activity tested...” becomes:

Figure 6.Antibacterial activity tested on agar paper disk diffusion (a) Staphylococcus aureus, (b) Escherichia coli and agar well diffusion (c) Staphylococcus aureus, (d) Eschiria coli.

Line 451: The word "who" should be "which".

Response:   Thank you, Line 451 was revised.

Reviewer 3 Report

Although the title "Chemical and structure assessment of new dental composites 2 with graphene exposed to staining agents" and the analysis are sound, there are still many areas that need to be revised.

Abstract-

Line 22- Present research uses small amounts of GO to enhance….. : A specific abstract experimental design of the recommended amount is required and also involves the following results need to be addressed.

Line 30- SEM microscopy: The abbreviation used in the abstract is inappropriate, please modify. The Abstract needs extensive revision to address important issues and the finding results in this study.

Introduction-

Lines 56-58 - Colour change of composite restorations in different colour…..: Please provide some reference here to explain this issue.

Materials and methods

2.1. lines 84-94: Please give the particle size distribution of all powders used in this study, as the resulting effects are very important.

Line 89- GO/ZrO2 was synthetized by mixing two suspensions:….: Since the author mentions the synthesis procedure, does this mean that GO reacts with ZrO2. please explain it.

Table 1: The amounts given here are vague, please give specific and individual amounts of each material that makes up the composite.

Line 118: please explain Ha-F

Lines 122-123 - particles were modified by using γ-methacriloyloxypropyl-trymethoxysilane (A-174) 122 (Sigma Aldrich, Germany)…..: Suggest an explanation and some analysis about the filler surface modifications, such as IR.

2.3. Lines 132-136: Can the authors provide images of the sample preparation for better understanding by the readers.

Line 144: Correct spelling.

Results- Please give n values for the number of replicates and show significant p-values to mark group differences in all plots.

A Discussion section is suggested as the results are not compared to any reference results and some discussions are made in their findings.

Author Response

Dear reviewer, 

The authors consider the reviewer’s comments and suggestions of highly scientific importance. Your ideas really helped us to improve the quality of our manuscript. We have corrected the errors that appeared in the manuscript and incorporated the changes in the revised manuscript using track changes.

Line 22- Present research uses small amounts of GO to enhance….. : A specific abstract experimental design of the recommended amount is required and also involves the following results need to be addressed.

Line 30- SEM microscopy: The abbreviation used in the abstract is inappropriate, please modify. The Abstract needs extensive revision to address important issues and the finding results in this study.

Response: We modified the abstract.

Introduction-

Lines 56-58 - Colour change of composite restorations in different colour…..: Please provide some reference here to explain this issue.

Response: We added new references [11,12,13].

 Materials and methods

2.1. lines 84-94: Please give the particle size distribution of all powders used in this study, as the resulting effects are very important.

Response: We added.

Line 89- GO/ZrO2 was synthetized by mixing two suspensions:….: Since the author mentions the synthesis procedure, does this mean that GO reacts with ZrO2. please explain it.

Response : Graphene consists in a one-atom-thick planar sheet of sp2 – bonded carbon atoms arranged in a hexagonal lattice. It is considered to be the “thinnest and strongest material in the universe” and it has remarkable physical and chemical properties, including superior Young’s modulus (1 TPa) and tensile strength (130 GPa). oxide (GO), as well as reduced graphene oxide (rGO) can be easily functionalized due to their abundance in oxygen containing groups (epoxy, hydroxyl and carboxyl) which enabled their use in a series of nanocomposites of different polymers with a wide range of applications. Among them, ZrO2 nanoparticles are one of the most attractive candidate that can be used to attach on the surface of GO due to their excellent hardness, mechanical strength, chemical and thermal stability or wear resistance.

 Table 1: The amounts given here are vague, please give specific and individual amounts of each material that makes up the composite.

 Line 118: please explain Ha-F

Response : In the composition isn’t present HA-F filler. It was a drafting error. Sorry for that.

Lines 122-123 - particles were modified by using γ-methacriloyloxypropyl-trymethoxysilane (A-174) 122 (Sigma Aldrich, Germany)…..: Suggest an explanation and some analysis about the filler surface modifications, such as IR.

Response: We added.

 2.3. Lines 132-136: Can the authors provide images of the sample preparation for better understanding by the readers.

Response:  We don't have, so sorry. We consider that is enough information in the literature about sample preparation. The sample preparations were made according to ADA Specification No. 27-1993 / ISO 4049/2000. 

Line 144: Correct spelling.

Response : We corrected.

Results- Please give n values for the number of replicates and show significant p-values to mark group differences in all plots.

Response: We added.

A Discussion section is suggested as the results are not compared to any reference results and some discussions are made in their findings.

Response : We added.

Round 2

Reviewer 1 Report

Abstract:

Specify (in parentheses) the abbreviations CC, GS, GZ

Inroduction:

The phrase “The novelty of this study consists in the fillers used in the composition of dental  composites to confer colour stability, and antibacterial properties” should be moved to the end of the Introduction section.

Reviewer 3 Report

The authors improved the manuscript to a great extent and successfully revised the manuscript. Therefore, I suggest it's acceptable.